# Regional Multi-ship Collision Risk Analysis Based on Velocity Obstacle Method: a Case Study on the Pearl River Estuary

Qi Liu
*School of Navigation*
*Wuhan University of Technology*
Wuhan, China
State Key Laboratory of
Maritime Technology and Safety,
Wuhan, China
lq754001x@whut.edu.cn

Pengfei Chen
*School of Navigation*
*Wuhan University of Technology*
Wuhan, China
State Key Laboratory of
Maritime Technology and Safety,
Wuhan, China
Chenpf@whut.edu.cn

Junmin Mou
*School of Navigation*
*Wuhan University of Technology*
Wuhan, China
State Key Laboratory of
Maritime Technology and Safety,
Wuhan, China
Moujm@whut.edu.cn

Linying Chen
*School of Navigation*
*Wuhan University of Technology*
Wuhan, China
State Key Laboratory of
Maritime Technology and Safety,
Wuhan, China
LinyingChen@whut.edu.cn

*Abstract*—**Analysis of regional multi-ship collision risk is essential for enhancing the efficiency of traffic management in maritime transportation. However, traditional collision risk analysis methods only assess the risk of collision from the viewpoint of ship pair encounters. In this research, a novel framework for analyzing regional multi-ship collision risk based on Velocity Obstacle (VO) method is proposed using the AIS (Automatic Identification System) data. Firstly, the ships in specific sea areas are clustered with the density-based spatial clustering of applications with Noise to identify multi-ship encounter situations. Afterward, a new collision risk indicator utilizing VO-based time-varying collision risk measurement method is proposed to calculate the collision risk of the single ship. Secondly, the macro-regional collision risk is quantified by calculating the contribution of each ship and each cluster with the Shapley value in cooperative games. Finally, to verify the effectiveness of the proposed framework, we carried out a case study of the Pearl River Estuary in China using historical AIS data. The results show that the proposed framework for regional multi-ship collision risk analysis can help maritime surveillance operators identify the ships with high risk and gain a better understanding of regional collision risk from both microcosmic and macroscopic perspectives.**

*Keywords—multi-ship encounter situation, velocity obstacle, time-varying risk, shapely value, maritime traffic safety*

## I. INTRODUCTION

Maritime transportation is one of the most important transportation approaches for international trade today. With the trend towards economic globalization, maritime transportation has continued to grow over the past decades. However, the increase in maritime transportation volume has led to the augment of maritime traffic density and maritime traffic complexity, thereby increasing the occurrence rate of maritime accidents, in particular ship collisions [1]. In the face of the relatively high maritime traffic volumes or complexity at sea, maritime surveillance operators are always subjective and random in the conduct of monitoring, lacking an overall perception of regional collision risk, regional collision risk, which brings large pressure on maritime surveillance. To analyse the risk of ship collision from multiple perspectives and enhance the regulatory efficiency of Vessel Traffic Service Operators (VTSO), it is imperative to put forward a novel framework for analysing regional multi-ship collision risk.

The analysis of ship collision risk is a research hotpot in the maritime field and plays an important role in reducing the number of collision accidents and enhancing the efficiency and level of maritime traffic monitoring. To assess the risk of ship collision quantitatively from multiple perspectives, many scholars have conducted plenty of research and proposed a variety of methods. These methods can be broadly categorized into three general groups: (1) synthetic indicator-based approaches; (2) safety domain-based approaches; and (3) velocity obstacles-based approaches.

Synthetic indicator-based approaches integrate some factors that indicate the spatial and temporal motion characteristics of encountering ships to measure the Collision Risk Index (CRI) using mathematical functions. The two most famous factors are Distance to Closet Point of Approach (DCPA) and Time to Closest Point of Approach (TCPA), which have been applied in [2-6]. In addition, Zhang et al. [7] considering some risk-influencing elements, introduced a new risk indicator named Vessel Collision Risk Operator (VCRO) to measure the level of the conflict risk of ships. Relevant work can further refer [8]. Based on Zhang et al research, [9] improved the relative distance in the original VCRO and proposed the model of enhanced vessel conflict ranking operator, which further enhanced the accuracy of conflict risk measurement.

Safety domain-based approaches usually construct the own ship's (OS) safety domain in space, take the ships

The work presented in this study is financially supported by the National Natural Science Foundation of China (Grant Number: 52101402, 52271367)

intruding into the safety domain of the OS as posing a collision risk, detect potential collision conflicts, and assess the risk of ship collision in terms of invasions or overlaps in the safety domain of encountering ships, such as ship domain [10] and collision diameter [11]. The ship domain has received a great deal of attention in recent years, and massive AIS (Automatic Identification System) data and advances in intelligent technologies have facilitated the development of various ship domain models with different shapes, including circular, elliptical [12], and polygonal [13]. These ship domains have been applied in collision risk analysis. For instance, Wang et al. [14] based on the elliptical ship domain, developed the Quaternion Ship Domain (QSD) by combining the impact of the COLREGS on the process of the actual ship encounter situations and used it for the assessment of ship collision risk. Szlapczynski et al. [15] developed the domain intrusion time/degree indicators to evaluate the collision risk collision during ship navigation. Liu et al. [16] proposed a collision probability model by introducing the maximum interval and the violation degree of two ship domains to measure the collision risk. Li et al. [17] proposed a novel collision risk assessment model based on the integration of elliptic and quadratic ship domains, offering a new way for collision risk measurement.

Velocity obstacle-based approaches transform the spatial-temporal correlations between ships into the velocity domain and judge the OS's velocity sets of falling into the dangerous velocity space to determine whether the collision risk exists. Recently, it has been progressively developed to combine the ship domain with the VO, proposing non-linear VO [18] and generalized VO algorithms [19], and the VO algorithms have been widely applied in ship collision risk analysis. For instance, Huang et al.[20] first developed VO-based Time-varying Collision Risk (TCR) measurement method to estimate the collision risk of the single ship in multi-ship encounters. Chen et al. [21] based on TCR measurement, introduced a real-time regional ship collision risk analysis method in different encounter situations. Li et al. [22] proposed a rule-aware TCR model for real-time collision risk analysis, which integrates the impact of various factors in the actual situation.

The above approaches provide a solid foundation for the development of collision risk analysis methods. However, these approaches mainly assess the collision risk from ship-ship encounters' viewpoint and analyses ship collision risk only from a microscopic perspective. With the gradual increase in the number of ships, multi-ship encounters are common at sea. Therefore, it is necessary to propose a novel framework to analyse the ship collision risk in the case of regional multi-ship encounters from multiple perspectives. Relevant work has been done. Zhang et al. [23] combined the density complexity and the multi-vessel collision risk operator to analyse regional vessel collision risk. Zhen et al. [24] considering the impact factors of DCPA, TCPA, ship crossing angle, and navigational environment, proposed a fuzzy logic-based collision risk model for regional multi-ship collision risk assessment. Besides, Liu et al. [25] developed a framework for regional collision risk identification with the spatial clustering method. The contribution of this study is to introduce a novel regional collision risk analysis framework that combines the TCR-based collision risk measurement and the Shapley value method. This framework can accurately identify high-risk ships and quantify the regional collision risk from both micro and macro perspectives, which will help the VTSO to accurately grasp the trend of the regional collision risk and strengthen their capacity and efficiency of maritime safety surveillance.

The structures of this paper are organized as follows. The methodology of the research is introduced in section II. Section III describes the construction of the framework. Section IV conducts a case study with the proposed framework for regional collision risk analysis. Some discussion about the results and comparison are presented to validate the effectiveness and feasibility of the proposed framework in section V. Finally, section VI concludes the research.

## II. METHODOLOGY

### A. Overview of the Study

In this study, the collision risk is defined as the percentage of velocities that might potentially result in a collision accident within the entire velocity sets of the OS. This definition comprehensively considers the motion state that the ship needs to maintain for effective collision avoidance from the free space's viewpoint and provides a quantitative measurement of the collision risk faced by the OS, which could significantly assist the VTSO in assessing and mitigating potential collision scenarios. Building upon this definition, we proposed a novel framework based on VO method to analyse regional multi-ship collision risk from both microscopic and macroscopic perspectives, which is beneficial to have an overall understanding of regional multi-ship collision risk and improve the efficiency of safety management for the VTSO in jurisdictional waters.

Firstly, the AIS data in the designated region will be collected and preprocessed over a specified time interval. Subsequently, the Density-Based Spatial Clustering of Applications with Noise (DBSCAN) method will be employed to classify the ships into different clusters. This density-based clustering technique takes into account the spatial distances between ships to identify regional multi-ship encounter situations, which are critical for effective analysis. Secondly, we utilize the TCR-based collision risk measurement to accurately quantify the collision risk for individual ships. Besides, by combining the Shapely value method, the collision risk of each cluster is measured by calculating the contribution of ships in a cluster. In this way, the macro-regional collision risk can be derived using the collision risk and the contribution of each cluster. Finally, to validate the effectiveness and feasibility of the proposed framework, we conduct two comparative experiments with the existing collision risk approaches. These experiments are designed to rigorously verify the performance of our framework against the traditional approach, allowing us to demonstrate its advantages in terms of accuracy and application in real sailing scenarios. The proposed research framework is shown in Figure 1.

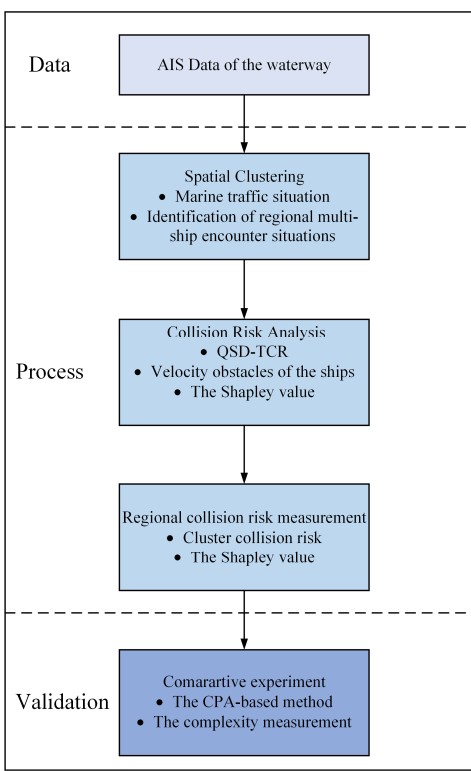

Fig. 1.  The proposed research framework

## B. Regional Multi-ship Encounter Situation Recognition Using Density-based Clustering

The density-based spatial clustering approaches are a fundamental category of unsupervised learning algorithms that have achieved widespread application in various applications due to their intuitive and fast advantages recently, mainly including the DBSCAN, hierarchical-DBSCAN, and Ordering Points To Identify the Clustering Structure. These methods are based on the principle that the spatial density distribution of the data is processed with a predetermined threshold to divide them into different groups [21]. In this research, we specifically utilise the DBSCAN method to conduct the clustering technology for the recognition of regional multi-ship encounter situations. This algorithm can divide similar data into the same cluster according to certain principles and find out the noise data that does not belong to any cluster. The implementation of the DBSCAN algorithm requires the setting of two primary parameters: Eps and MinPts. The pseudocode for the DBSCAN algorithm is described in Figure 2. By employing the DBSCAN method, the ships in a selected region can be classified into multiple clusters, which can reduce the burden of collision risk calculation and improve the efficiency of recognizing muti-ship encounter situations. During the clustering process, the ships that are not included in any clusters can be considered noise points in the clustering process. These noise points are spatially distant from other vessels and are considered to have no collision risk with others. Therefore, we can disregard these ships in this research, which can help simplify the calculation of collision risk.

| Algorithm 1: The implementation process of DBSCAN algorithm |
| --- |
| Input: |
|     *D*: a dataset contain n objects |
|     *Eps*: neighborhood parameter |
|     *MinPts*: neighborhood parameter |
| Output: a set of  clusters |
| Start: |
|     1. Mark all objects as unvisited |
|     2. DO |
|     3. Randomly select an unvisited object *p*; |
|     4. Mark *p* for visit; |
|     5. If *p* has least *MinPts* in its *Eps* neighborhood; |
|     6.    Create a new cluster C and add *p* to C; |
|     7.    Sets of objects in *Eps* neighborhoods where *N* is *p*; |
|     8.    For each point in *Np* |
|     9.    If *p\** is un visited; |
|     10.    Mark  *p\** for visit; |
|     11.    If *p\** has at least *MinPts* objects in its *Eps* neighborhood, add them to *N*; |
|     12.    If *p\** is not a member of any cluster, add *p\** to C; |
|     13.    End for; |
|     14.    Output C; |
|     15. Else marker *p* is noise; |
|     16. Until there is no object marked unvisited; |
| End |

Fig. 2.  The pseudocode for the DBSCAN algorithm

## C. TCR-based Multi-ship Collision Risk Measurement Model

Traditional collision risk analysis methods approaches consider the spatiotemporal relationships of encounter ships separately, which can bring contradictory results. To overcome this shortcoming, the TCR collision risk modeling method is employed in this research to analyse and quantify the risk of ship collision. The concept of TCR, first proposed by [20], is described as the likelihood of the event that the OS will not be able to avoid a collision with other ships. The TCR for the collision risk measurement projects the spatiotemporal relationships between ships in the OS's velocity space and assesses the difficulty of avoiding collision accidents. The description of TCR is shown in (1) and Figure 3.

$$TCR(t) = \frac{sets_{collision}(t)}{sets_{reachable}(t)} \tag{1}$$

where $sets_{collision}(t)$ are the sets of velocities that lead to collisions at time $t$; $sets_{reachable}(t)$ are the OS's reachable velocities sets before collisions at time $t$.

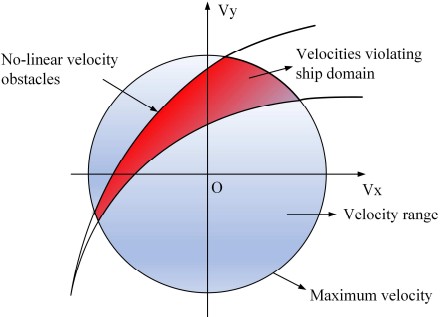

Fig. 3.  The description of TCR

## D. Shapley Value Method in Cooperative Games

Cooperative games involve competition between different groups that need both coalition and cooperation. It is used to ascertain how to distribute the amounts produced by cooperation, which can be used to measure the contribution of the individual in the group [26]. The Shapley value method, introduced by Shapley and Shubik in 1953 [27], plays a dominant role in cooperative game theory. It allocates

cooperative amounts by estimating the contribution of each player. The formula of the Shapley value method is shown as (2):

$$SV_i[A] = \sum_{\substack{C \subseteq N \\ i \subseteq C}} \frac{(c-1)!(n-c)!}{n!} [A(C) - A(C - \{i\})] \quad (2)$$

where $i$ is the player in the game, $C$ signifies the group generated by the player $i$, $c$ represents the total number of players of the group $C$. $N$ denotes the group formed by all vessels, $n$ denotes the number of players in group $N$. $A(C)$ refers to the amounts generated by the group $C$, $A(C - \{i\})$ refers to the amounts generated by group $C$ before player $i$ joins. $SV_i[A]$ represents the Shapley value of the player $i$.

The Shapley value method was first applied in the maritime field to assess the contribution of ships to the global collision risk [26]. In this study, the Shapley value method is also employed to identify the contribution of each ship and cluster to the regional collision risk. With this indicator, the measurement of regional collision risk from a macroscopic viewpoint can be obtained.

### III. THE CONSTRUCTION OF FRAMEWORK

#### A. Analysing the Risk of Ship Collision in Multi-ship Encountering

The role of the TCR method could be to detect the collision candidate ships and provide the measurement of collision risk for the single ship navigating at different sea areas. Considering these advantages, we utilize the TCR-based collision risk modeling method to analyse the collision risk of ships in this paper.

The VO method can collect some velocity sets that could lead to collisions between the OS and the TSs, which is essential for the TCR. Supposing that ship $A$ and ship $B$ navigate in the waterways. The motion status of the two ships can be denoted as $A\{P_A(T), V_A(T), L_A\}$, $B\{P_B(T), V_B(T), L_B\}$; $P$ is the position of two ships at time $T$. $V$ is their velocity at time $T$. $L$ is the length of the ships. Using the VO method, the spatiotemporal correlations between two ships are transformed into the ship A's velocity space. The condition of collision can be shown as (3).

$$P_A(t_c) \subseteq P_B(t_c) \oplus ConfP$$
$$P = P(t_0) + v*(t - t_0) \quad (3)$$

where $P_A(t_c)$, $P_B(t_c)$ refers to the position of ship $A$ and ship $B$ at collision time $t_c$; $P$ is the position of two ships at the specified time $t$. $ConfP$ are all the possible positions of ship $A$ around ship $B$ when the collision happens. $\oplus$ represents the Minkowski addition.

In this research, we utilize the NLVO method to obtain the VOs in TCR. The NLVO method can be expressed in (4):

$$NLVO_{A|ship_{jti}} = \bigcup_{t_f}^{\infty} (\frac{P_{shipj}(t_i) - P_A(t_0)}{(t_i - t_0)}) \oplus \frac{ConfP_{ship_j}}{(t_i - t_0)} \quad (4)$$

$$NLVO_{A|allship_{ti}} = \bigcup_{j=1}^{n} NLVO_{A|ship_{jti}}$$

where $P_{shipj}(t_i) - P_A(t_0)$ indicates the difference in distance between the ship $j$ at the time $t_i$ and the OS at time $t_0$. $NLVO_{A|ship_{jti}}$ denotes the OS's velocity sets induced by ship $j$. $NLVO_{A|allship_{ti}}$ denotes the OS's velocity sets induced by all target ships based on Boolean operations. To take full account of the ship's maneuverability, velocity, and heading influences, we employ the QSD as a criterion for $ConfP$. A detailed description of the QSD can be found at [28, 29].

To quantify the collision risk of the individual ship, a new collision risk indicator-$TCR_{QSD}$, which is the TCR measured by the OS's QSD, is introduced in this study. The calculation formula of the indicator is shown in (5):

$$TCR_{QSD} = \frac{VO_{QSD}}{VO_{region}} \quad (5)$$

where $VO_{QSD}$ is the area of intersection regions between the VOs induced by the QSD of TS and the velocity region of the OS. $VO_{region}$ is the area of the ship's velocity region, representing all the possible velocities that the ship can achieve. To simplify the calculation process, the assumption that the changes of course and reduction of velocity are considered collision avoidance operations to obtain the ship's velocity region. Using this indicator, the measurement of the collision risk of single ships can be proceeded.

#### B. Identifying the Contribution of Each Ship to the Regional Collision Risk in Multi-ship Encountering

The Shapley value method can measure the contribution of players to the entire group mentioned in section II. Inspired by this research, the Shapley value method is employed in this paper to estimate the contribution of each ship and cluster to the regional collision risk.

At sea, the multi-ship encounters in a region can be considered cooperative games. The ship in a multi-ship encountering situation can be considered as a game player and the numerical values collision risk of the ship is equivalent to the amount made by the game player. The ships are arranged in the way of permutation and combination to produce various groups. The amount of collision risk for each ship group $A(C)$ should first be obtained. The amount of group is regarded as the sum of the collision risk of each ship in a multi-ship encounters group. Besides, $A(C - \{i\})$ could be obtained by calculating the amounts of the collision risk of ship group $C$ without the participation of ship $i$. Finally, each ship's Shapley value could be measured based on (2). Combining the collision risk values for individual ships, the collision risk of clusters can be obtained based on (6). In this way, the regional collision risk from a macroscopic perspective can be also quantified based on (7).

$$CCR_i = \bigcup_{i=1}^{n} (TCR_{QSD} * S_i) \quad (6)$$

$$M - RCR = \bigcup_{j=1}^{m} CCR_j * S_j \quad (7)$$

where $TCR_{QSD}$ is the numerical value collision risk for single ships. $CCR_i$ denotes the collision risk of each cluster. $M - RCR$ refers to the macro-regional collision risk. $S_i$ and $S_j$ denotes the Shapley value of each ship and each cluster, respectively. $n$ represents the number of ships in a cluster. $m$ represents the number of clusters in the research region.

## IV. CASE STUDY

To validate the feasibility of the proposed framework, in this section, we carried out a case study on the Pearl River Estuary in China for regional multi-ship collision risk analysis. The elaboration of research data and detailed results of the experiment are shown in the following section.

### A. Description of the AIS Data and Parameter Setting

In this study, we used the Pearl River Estuary's AIS dataset for one day, which was provided by the Wuhan University of Technology. The AIS data, showing the mooring and berthing status of the vessel, were removed to avoid the influence of abnormal data on the case study. Besides, the ship type was not considered in this research. The MinPt and Eps in the DBSCAN algorithm are set to 2 and 6 nm, respectively. The detailed parameter settings are displayed in Table I.

TABLE I. PARAMETER SETTINGS

| Variable | Setting |
|---|---|
| Time | 08:00 15th - 08:00 16th May 2020 |
| Data boundary | Lat: 21.7410-22.1289°N; Log:113.2370-113.7677E; |
| Eps | 6nm |
| MinPt | 2 |
| TCR time | 30min |
| Ship Length (if data not available) | 200m |

### B. Results of The Experiments

In this section, we randomly selected two sets of ship AIS data at different moments to validate the effectiveness of the proposed TCR-based multi-ship collision risk analysis framework. The ships in the designated region are divided into different clusters using the DBSCAN algorithm, then the indicator of the QSD-based TCR that represents the collision risk for single ships, and the M-RCR can be obtained utilizing the proposed framework. Figures 4 and 5 show the visualization of the ship clustering and randomly selected ships' TCR at different moments. The detailed experimental results for these ships are illustrated in Tables II and III.

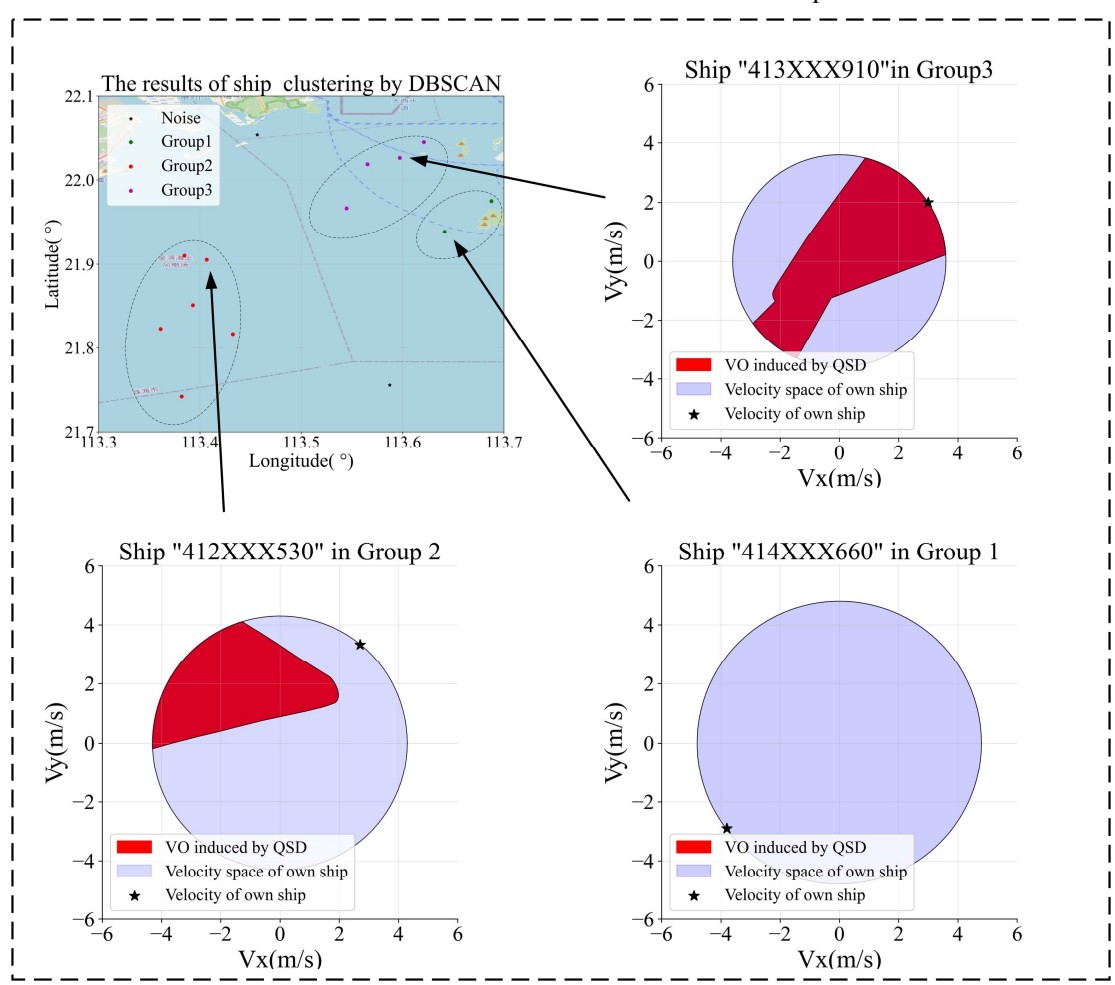

Fig. 4.   Visualization of ship clustering and the ships' TCR in different groups at 13:25:00

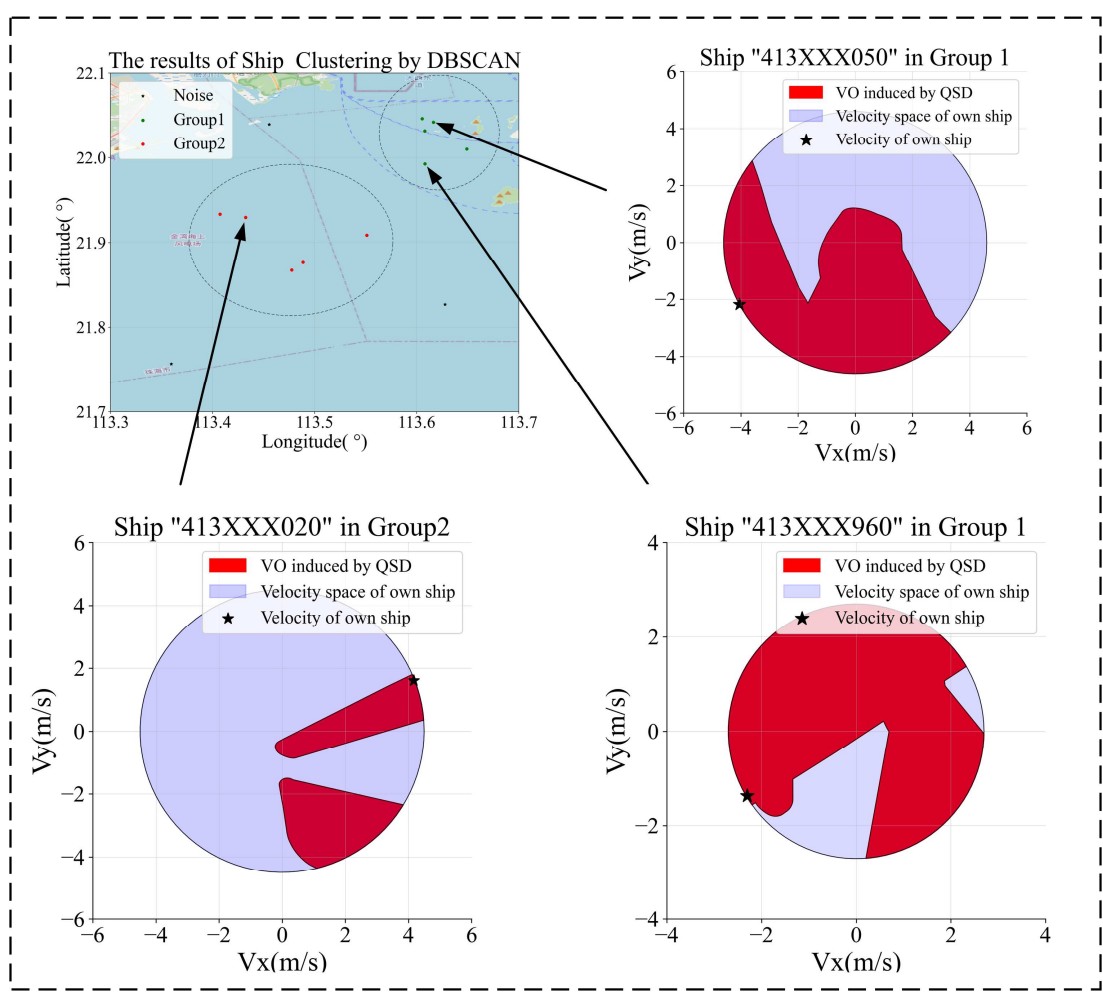

Fig. 5.  Visualization of ship clustering and the ships' TCR in different groups at 21:30:00

From Figures 4 and 5, it can be found that more than 10 ships are navigating in the research region at both timespots. For timespot 13:25:00, there were 15 ships in the region, which were categorized into three ship groups by implementing the DBSCAN method. To demonstrate the performance of the proposed collision risk analysis framework, the TCR has been visualized for three ships (414XXX660, 412XXX530, 413XXX910). For ship "414XXX660" contained in Group 1 (green), the two ships in Group 1 did not form an encounter situation since the trajectories of the two ships observed from AIS data are divergent. Therefore, there is no collision risk between the two ships, and the QSD-TCR of the ship "414XXX660" is 0. Meanwhile, ship "412XXX530" in Group 2 (red) had formed the encounter situation with one ship of the group. The QSD-TCR of the ship " 412XXX530" is 0.2625, which shows less collision risk. Differing from ship "414XXX660" and ship "412XXX530", ship "413XXX910" in Group 3 (purple) had formed multiple encounter situations with the rest of the ships in the group, thus ship "413XXX910" has a higher collision risk (QSD-TCR:0.4295) than two ships. In addition, "noises" are successfully recognized by the DBSCAN algorithm.

For timespot 21:30:00, the experimental results for the ships are available utilizing the proposed framework. These

ships in the region are classified into two clusters with the DBSCAN method, and each cluster contains five ships. The TCR has also been visualized for three ships (413XXX050, 413XXX960, 413XXX020), For ship "413XXX050" and ship "413XXX960" in Group 1 (green), the two ships had formed encounter situations with several ships of Group 1, showing a high collision risk. The QSD-TCR of ship "413XXX050" and "413XXX960" are 0.4788 and 0.7944, respectively. Based on the proposed collision risk analysis framework, the two ships should take immediate collision avoidance measures to mitigate the collision risk at the moment. Besides, ship "413XXX020" in Group 2 (red) also formed encounter situations with several ships. However, the values of QSD-TCR of the ship "413XXX020" are small. The reason is that the distance between ship"413XXX020" and the ships forming the encounter situation is relatively large. However, a collision accident could occur if the ship "413XXX020" continues to navigate in its current state of motion. The ship "413XXX020" should take collision avoidance operations as far as possible. Besides, each ship's Shapely value can be calculated utilizing the proposed framework. The detailed results of the case study are shown in Table II.

Finally, with the Shapley value indicating the contribution of each ship and cluster, the numerical values of M-RCR at

both timespots 13:25 and 21:30 can be measured in the selected region, as shown in Table III. Comparing the M-RCR at timespot 13:25, the values of M-RCR at timespot 21:30 are higher. The VTSO should devote more effort to strengthening the supervision and management of the region at the moment, which can help them accurately grasp the trend of the collision risk from a macroscopic perspective. In conclusion, the proposed collision risk analysis framework can detect ships with high risk and quantify the temporal and spatial distribution of collision risk in designated regions. The VTSO can take action to enhance the supervision of the maritime traffic situation to ensure the safety of ship navigation.

TABLE II. THE COLLISION RISK VALUE FOR SINGLE SHIP UTILIZING THE PROPOSED FRAMEWORK

| Time | MMSI | $TCR_{QSD}$ | Shapely value | Group |
|------|------|------|------|------|
| 13:25:00 | 413XXX910 | 0.4295 | 0.2803 | 3 |
| 13:25:00 | 412XXX530 | 0.2625 | 0.1948 | 2 |
| 13:25:00 | 414XXX660 | 0 | 0 | 1 |
| 21:30:00 | 413XXX050 | 0.4788 | 0.2634 | 1 |
| 21:30:00 | 413XXX960 | 0.7944 | 0.3818 | 1 |
| 21:30:00 | 413XXX020 | 0.1871 | 0.1524 | 2 |

TABLE III. THE RESULTS OF MACRO-REGIONAL COLLISION RISK UTILIZING THE PROPOSED FRAMEWORK

| Time | M-RCR |
|------|------|
| 13:25:00 | 0.2739 |
| 21:30:00 | 0.6405 |

## V. DISCUSSION

In the previous sections, multi-ship encounter situations are identified, and the collision risk of the single ship and the regional collision risk are analysed and quantified. To further validate the effectiveness of the proposed collision risk analysis framework. In this section, two comparative experiments employing the traditional collision analysis methods proposed by [25, 30] will proceed: (1) a comparison between the proposed framework and the CPA-based method [25]. (2) a comparison between the proposed framework and the complexity measurement-based method [30]. The comparison has two parts, mainly including the comparison between the collision risk and complexity of a single ship and the results of regional collision risk and overall complexity of the selected region. The results are shown in Tables IV and V.

TABLE IV. RESULTS OF COLLISION RISK ANALYSIS AND COMPLEXITY OF SHIP UTILIZING THE METHODS [25, 30]

| Time | MMSI | $TCR_{QSD}$ | CRI | Complexity | Group |
|------|------|------|------|------|------|
| 13:25:00 | 413XXX910 | 0.4295 | 0.3699 | 4.3364 | 3 |
| 13:25:00 | 412XXX530 | 0.2625 | 0.2778 | 0.7242 | 2 |
| 13:25:00 | 413XXX660 | 0 | 0 | <0.0001 | 1 |
| 21:30:00 | 413XXX050 | 0.4788 | 0.4290 | 8.2786 | 1 |
| 21:30:00 | 413XXX960 | 0.7944 | 0.5251 | 6.8122 | 1 |
| 21:30:00 | 413XXX020 | 0.1871 | 0.3668 | 1.8870 | 2 |

TABLE V. RESULTS OF REGIONAL COLLISION RISK (RCR) AND COMPLEXITY IN REGION UTILIZING THE METHODS [25, 30]

| Time | M-RCR | RCR | Complexity |
|------|------|------|------|
| 13:25:00 | 0.2739 | 0.3472 | 0.3012 |
| 21:30:00 | 0.6405 | 0.5654 | 6.2923 |

In Tables IV and V, it can be seen that although the numerical values for single-ship collision risk (QSD-TCR, CRI) derived from the proposed algorithm and the CPA-based method for the same scenario have differences. The final results, which indicate the high-risk ships, are consistent with each other. Meanwhile, the region collision risk (M-RCR, RCR) from the two different methods also yields different results, but the region with high risk identified by the CPA-based method is in line with the proposed algorithm. These verify the performance of the proposed framework to identify ships with high collision risk from a microcosmic perspective and gain an overall collision risk in a region from a macroscopic perspective. In addition, The traffic complexity model, first proposed by [30], is used to assess the complexity of maritime traffic situations. According to [31], there is a certain correlation between traffic complexity and the risk of ship collision. In general, the higher the traffic complexity, the greater the risk of collision. It can reflect the magnitude of the instantaneous ship collision risk. Therefore, we introduce this indicator as a comparison to further verify the effectiveness of the proposed framework. From Tables IV and V, the traffic complexity of ships obtained from the complexity measurement-based method can also identify the ships and regions with high risk, and the results are consistent with the proposed framework. In conclusion, we further validate the effectiveness and feasibility of the proposed framework in analysing the collision risk under the multi-ship encounter situations in the region by the two comparative experiments.

## VI. CONCLUSION

In this paper, a novel regional multi-ship collision risk analysis framework based on the VO method is proposed. The risk of collision is described as the proportion of the velocity obstacle sets generated by the TSs. One risk indicator $TCR_{QSD}$ is introduced to quantify the collision risk for single ships. Combining the Shapley value method in cooperative games, the macro-regional collision risk can be obtained.

A case study on the Pearl River Estuary to validate the feasibility of the proposed framework. The results indicate that the proposed framework can accurately identify high-risk ships and regions. Comparing the existing collision risk analysis method, the proposed framework can analyse the collision risk of multi-ship encounter situations in a region from both micro and macro perspectives. The contribution of the proposed framework is that utilising spatial clustering techniques and the VO method apply them in the monitoring and management of collision risk in waters under the jurisdiction of maritime authorities. Based on this, maritime surveillance operators can have a better understanding of regional collision risk in multi-ship encounters, thus further enhancing the situational awareness ability and improving the efficiency of maritime traffic management faced with relatively high maritime traffic volumes or complexity challenges. However, the currently proposed framework has some shortcomings. One shortcoming is that the influence of ship heading angle is not considered in the clustering classification, which may affect the accuracy of multi-ship encounter recognition. In addition, other risk influencing factors (eg: weather conditions and ship type) are not integrated into the collision risk model. Future work could

focus on improving these limitations and applying the proposed framework to predict the regional collision risk.

## ACKNOWLEDGMENT

This work is financially supported by the National Natural Science Foundation of China under grants 52101402 and 52271367.

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
