# OpenReview forum: "Regional Multi-ship Collision Risk Analysis Based  on Velocity Obstacle Method：a Case Study on the  Pearl River Estuary"
_IEEE.org/ICIST/2024/Conference — IEEE ICIST 2024 Conference Submission_

### Official Review · Reviewer_VwR1 · 2024-08-25
**minor repair**

**Rating:** 8
**Confidence:** 3

**Review:**

1. The contributions of this manuscript need to be clearly presented and should be rephrased.
2. There are many symbols involved in the text, such as equation (3). It is recommended that the author explain and clarify them for readers to understand.

---

### Official Review · Reviewer_WM4Z · 2024-09-02
**This paper can be accepted.**

**Rating:** 7
**Confidence:** 5

**Review:**

This paper proposes a novel framework for analysing regional multi-ship collision risk based on Velocity Obstacle method.The reviewer's comments are as follows:
1.The English grammar and format of this manuscript could be further polished and checked carefully.
2.The format of references is not uniform.
3.To enhance the quality of the manuscript, it is recommened refining the language to improve readability and ensuring that the concepts are communicated accurately.
4.In addition to the contribution of the proposed work, it is suggested to discuss its main limitations/shortcomings.

---

### Official Review · Reviewer_XgCL · 2024-09-03
**Regional Multi-ship Collision Risk Analysis Based on Velocity Obstacle Method：a Case Study on the Pearl River Estuary**

**Rating:** 7
**Confidence:** 4

**Review:**

1. The paper is well-structured and provides a comprehensive overview of the proposed framework for regional multi-ship collision risk analysis. However, some sections could benefit from further clarification to improve readability for a broader audience. For instance, a brief introduction to the concepts of DBSCAN, TCR, and Shapley value method at the beginning of the methodology section would help readers who are less familiar with these techniques. Additionally, the conclusion section could summarize the key findings and implications of the study in a more concise manner.
2.The current study focuses on collision risk analysis based on velocity obstacle and Shapley value methods. It would be interesting to explore potential extensions of the framework, such as incorporating real-time data streams for dynamic risk assessment or integrating other factors that may influence collision risk (e.g., weather conditions, ship types, and traffic patterns). Additionally, discussing potential applications of the framework in maritime traffic management and safety regulations could further highlight its practical value and impact.

---

### Decision · Program_Chairs · 2024-09-06

Accept (Oral)